# OpenReview forum: "Context Tuning for In-Context Optimization"
_ICLR.cc/2026/Conference — Submitted to ICLR 2026_

### Official Review · Reviewer_Qgpc · 2025-10-23

**Soundness:** 3
**Presentation:** 4
**Contribution:** 2
**Rating:** 4
**Confidence:** 4

**Summary:**

This paper introduces **context tuning**, a variant of existing continuous (i.e., parametric) prompt optimization approaches.
Unlike previous methods that typically initialize parameters randomly, the proposed approach initializes them using demonstration representations.
For training, it also incorporates additional techniques such as leave-one-out masking and token dropout, which resemble those used in the previous work called Test-Time Training (TTT).
When evaluated on several benchmarks, including NLP-LR, MMLU, BBH, and ARC, the proposed method achieves performance comparable to existing methods and further improves results when combined with TTT.

While the paper introduces a new paradigm termed **in-context optimization**, encompassing both the proposed method (i.e., context tuning) and TTT, it remains unclear whether this terminology provides genuine conceptual novelty or clarity. The idea can arguably be well captured by existing notions such as context engineering, prompt tuning, or test-time compute scaling.

**Strengths:**

- The draft is well-written and easy to understand, with clear notations and explanations.
- It is well-aligned with related work in the literature, presenting them within an integrated framework.
- The experiments cover a reasonable range of possibilities, addressing different tasks, models, and configurations.

**Weaknesses:**

- I appreciate the simplicity of the proposed idea, but it seems somewhat too incremental to merit a full-paper submission. This concern becomes particularly evident when compared with the previous work, TTT, where the only difference—at least as I understand it—lies in whether the model tunes LoRA adapters (possibly randomly initialized) or continuous prompts initialized with demonstration embeddings. Although ICLR does not offer a short-paper track, the contribution and scope of this work appear more suitable for a short-paper format in other conferences.
- There is no clear analysis of why context tuning achieves further gains compared to foundational approaches such as prompt tuning or prefix tuning. For instance, do the improvements stem from initialization strategies, leave-one-out masking, or other factors? While Table 3 presents ablation results, the connection between these variants and the original baseline remains ambiguous. A more detailed investigation of this aspect could reveal potential directions for further improvement.
- More fundamentally, it is intriguing that the trade-off inherent in these test-time training–like approaches can still be considered worthwhile, especially given their sensitivity to overfitting and their requirement for independent copies of task-specific model parameters—both of which contradict the philosophy of in-context learning and the recent paradigm of LLM usage. It is worth questioning whether, if we were to manually or automatically optimize our natural language prompts with an effort comparable to that required for such fine-tuning, we could achieve similar performance guarantees. I believe that the true merit of LLMs lies in their general capability to handle multiple tasks simultaneously within a single model, without the need for task-specific optimization once required for models such as BERT and its variants. From this perspective, it may be more appropriate to compare the performance of such approaches to that of traditional fine-tuning methods applied to encoder or decoder models, given their conceptual resemblance to fine-tuning in any case.

**Questions:**

Please see the Weaknesses section.

---

> ### Author Response · Authors · 2025-11-21
>
> We thank the reviewer for their thoughtful feedback. We are encouraged that our work is considered well written, aligned with related work, and supported by comprehensive experiments. We address the concerns point by point.
>
> > I appreciate the simplicity of the proposed idea, but it seems somewhat too incremental…
>
> We appreciate the reviewer’s recognition of the simplicity of Context Tuning, and believe that simplicity paired with consistent improvements makes it a practical adaptation strategy.
>
> **Context Tuning is substantially different from TTT in mechanism and efficiency.** While both fall under In-Context Optimization, TTT repeatedly reorders demonstration pairs, applies leave-one-out over token sequences, and updates LoRA parameters, requiring the model to reconstruct the task representation many times. In contrast, CT-KV extracts a structured KV representation from demonstrations once and directly optimizes this persistent representation, with Leave-One-Out Masking applied at KV segments rather than token sequences. This yields distinct optimization dynamics and the provable efficiency gains: Appendix B shows CT-KV’s training time scales linearly in demonstration counts, while TTT scales quadratically. Empirically (Table 1), CT-KV requires far less training time than TTT while it achieves competitive or better performance. These distinctions in mechanism and efficiency make Context Tuning meaningfully more than an incremental variant of TTT.
>
> > There is no clear analysis of why context tuning achieves further gains compared to foundational approaches…
>
> We appreciate the interest in deeper clarity on Context Tuning’s gains. In particular, Table 3 cleanly decomposes CT-KV’s gains over Prefix Tuning. Section 5.7 interprets these results, which we summarize into a row-by-row format:
> 1. **Neither.** This initialization-only variant resembles Prefix Tuning but replaces random vectors with demonstration-derived ones. It performs similarly to Prefix Tuning (Table 1), confirming that initialization alone is insufficient because the model retrieves targets in its initialized context.
> 2. **No Leave-One-Out Masking.** Adding Token Dropout provides mild regularization and marginal gains.
> 3. **No Token Dropout.** Leave-One-Out Masking alone yields a large improvement by preventing retrieval of target outputs and forcing generalization from other examples.
> 4. **Both.** Using both mechanisms yields the highest performance on NLP-LR, MMLU, and BBH (with ARC discussed in Section 5.7).
>
> > The idea can arguably be well captured by existing notions such as context engineering, prompt tuning, or test-time compute scaling.
>
> > More fundamentally, it is intriguing that the trade-off inherent in these test-time training–like approaches…
>
> We thank the reviewer for these broader questions on the positioning of test-time optimization relative to ICL. While ICL enables inference-time adaptation without further finetuning, we claim that ICO methods like TTT and Context Tuning introduce stronger task-specific adaptation. This is precisely why our evaluation includes both ICL and traditional finetuning baselines (Prompt/Prefix Tuning, LoRA variants), with Context Tuning showing significant gains over all of them (Table 1).
>
> **Conceptually, we view ICO as bridging ICL and finetuning.** Context Tuning leverages the LLM’s inherent ICL ability by initializing from demonstrations, while gradient-based optimization moves beyond superficial demonstration patterns [1]. This hybrid approach preserves the portability and safety of frozen-weight models while surpassing both ICL and finetuning baselines across benchmarks.
>
> **Task-specific prompts require manual engineering, while Context Tuning does not.** While well-engineered natural-language prompts could yield similar improvements, such prompt design requires significant manual, task-specific effort. In contrast, Context Tuning requires no manual engineering: across all 27 BBH tasks, a single hyperparameter setting is used, with Leave-One-Out Masking and Token Dropout helping to avoid overfitting. Hence CT provides automated, task-agnostic improvements where manual prompting requires substantial effort.
>
> Overall, Context Tuning achieves performance well beyond ICL and traditional finetuning, suggesting test-time optimization is a valuable complement to ICL. We thank the reviewer again for the thoughtful, broad questions.
>
> ### **Concluding Note**
>
> We have carefully addressed the concerns regarding contribution, ablation interpretation, and the broader position of test-time optimization. Given your recognition of the paper’s strong writing, alignment with related work, comprehensive experiments, as well as our strengthened explanations here, we kindly ask you to consider raising your score, as we believe the contributions will be valuable to the community.
>
> [1] Min et al., “Rethinking the role of demonstrations: What makes in-context learning work?”, 2022.

---

### Official Review · Reviewer_HFNj · 2025-11-01

**Soundness:** 3
**Presentation:** 2
**Contribution:** 2
**Rating:** 2
**Confidence:** 3

**Summary:**

This paper proposes Context Tuning, which initializes and optimizes a task’s few-shot demonstration-derived context, either a soft prompt (CT-Prompt) or per-layer KV prefixes (CT-KV), to boost adaptation without updating model weights. CT-KV conditions all layers through the KV cache and trains with linear cost in the number of demos (vs. quadratic for CT-Prompt and Test-Time Training), aided by leave-one-out masking and token-dropout. Across CrossFit, UnifiedQA, MMLU, BBH, and ARC, CT-KV outperforms ICL and classic prompt/prefix-tuning, rivals TTT, and combining TTT + CT-KV yields the best accuracy–efficiency trade-off. The work unifies these under an In-Context Optimization (ICO) view of updating either the model or its context.

**Strengths:**

1. **Clear and well-structured presentation.**
The paper is well-written and easy to follow. It clearly explains the intuition behind context optimization, the difference between CT-Prompt and CT-KV, and the rationale for efficiency gains. Figures and ablations effectively illustrate the role of leave-one-out masking and token dropout.
2. **Consistent and measurable improvement.**
The proposed CT-KV achieves clear and repeatable performance gains over standard ICL, prompt/prefix tuning, and even approaches or complements test-time training performance at a fraction of the cost. The results are consistent across multiple datasets and noise settings.

**Weaknesses:**

1. **Limited novelty.** The method conceptually extends test-time training by performing parameter-efficient adaptation with in-context examples, but mainly replaces LoRA with other PEFT methods such as prompt-tuning or prefix-tuning. As such, the contribution feels incremental rather than conceptually new. Moreover, there is prior work on few-shot prompt/prefix tuning since 2022 (e.g, studies exploring better initialization and adaptation strategies [1][2][3]), which are not discussed, weakening the positioning relative to earlier literature.

2. **Unclear model–task pairing.**
The choice of models and benchmarks appears somewhat arbitrary. Different tasks are paired with different models without clear justification, making cross-task comparisons difficult. Additionally, some benchmarks (e.g., NLP-LR) are dated and may not fully reflect the challenges of modern large-model evaluation.

3. **Limited generality and applicability.**
The method assumes explicit (x, y) demonstration pairs and does not naturally accommodate COT, which limits its applicability to more challenging tasks like math, or coding where intermediate steps matter. Since modern LLM use is increasingly zero-shot or instruction-driven, requiring curated few-shot pairs at test time may reduce its relevance to real-world interactive or open-ended scenarios.

[1] Pan et al, Self-supervised meta-prompt learning with meta-gradient regularization for few-shot generalization, 2023 \
[2] Huang et al, Learning a better initialization for soft prompts via meta-learning, 2022 \
[3] Qin et al, Learning to Initialize: Can Meta Learning Improve Cross-task Generalization in Prompt Tuning?, 2023

**Questions:**

See Weakness.

---

> ### Author Response · Authors · 2025-11-21
>
> We thank the reviewer for their thoughtful feedback. We are encouraged by the positive assessment of the paper’s clarity and recognition of Context Tuning’s strong performance across various evaluation settings. We thoroughly address the concerns below and revise our submission accordingly.
>
> > Limited novelty.
>
> Our goal is to situate Context Tuning within the broader landscape of few-shot adaptation methods. Relative to Prompt/Prefix Tuning, Context Tuning leverages in-context demonstration pairs to initialize the prompt/prefix, thereby directly exploiting the model’s ICL capability rather than optimizing randomly initialized vectors. Context Tuning is additionally distinct from LoRA-based approaches by not requiring weight updates. We designed the method to be simple to implement and broadly effective, which we demonstrate via evaluations across diverse benchmarks (Section 5.4), model scales from 1B to 32B (Section 5.4 and Section 5.6), and noisy conditions (Section 5.5).
>
> |  | LoRA | Prompt/Prefix Tuning | Context Tuning |
> |-|-|-|-|
> | Model Weight Updates | Yes | No | **No** |
> | In-Context Demonstration Examples | No | No | **Yes** |
>
> In Section 4.3, Context Tuning additionally introduces two key design components, Leave-One-Out Masking and Token Dropout, whose importance is validated by ablations in Table 3. These choices further differentiate Context Tuning from prior works.
>
> > Moreover, there is prior work on few-shot prompt/prefix tuning…
>
> We thank the reviewer for pointing out [1], [2], and [3]. These works indeed relate to Context Tuning at the level of learning prompts; we have added them to our Related Works section. However, they focus on meta-learning across many tasks, whereas Context Tuning performs single-task, inference-time adaptation. This distinction avoids the heavy computational and data requirements of multi-task meta-training. Moreover, as noted in Section 5.4, CT-KV surpasses MetaICL on NLP-LR, demonstrating that inference-time optimization on a single task can rival approaches that fine-tune model weights across many tasks. We also compare against more recent adaptation methods (e.g., DoRA, TTT) in Table 1, where Context Tuning remains superior.
>
> > Unclear model-task pairing.
>
> We thank the reviewer for raising this point of confusion. Our model-task choices are driven by computational feasibility and established baselines rather than arbitrary selection. As noted in Section 5.2:
> 1. GPT-2 for NLP-LR and Llama3-8B for BBH follow established baselines MetaICL and TTT, respectively.
> 2. To demonstrate that Context Tuning performs well across different model sizes beyond the two models above, we select Llama3-8B for MMLU.
> 3. We use Llama3-1B for ARC due to the high computational cost for processing its long sequences.
>
> We do not make cross-task comparisons because our goal is not to compare benchmark difficulties, but to evaluate Context Tuning across diverse settings, each paired with standard or computationally appropriate models. As further evidence of Context Tuning’s generalizability, Section 5.6 shows that CT-KV continues to outperform baselines when both are scaled to 32B models.
>
> Finally, while NLP-LR is our oldest benchmark, benchmarks like ARC and BBH are actively used in recent works [4, 5].
>
> [4] Hu et al., “Automated Design of Agentic Systems”, ICLR 2025
>
> [5] Teng et al., “Atom of Thoughts for Markov LLM Test-Time Scaling”, NeurIPS 2025

---

> > ### Author Response · Authors · 2025-11-21
> >
> > > Limited generality and applicability.
> >
> > **Context Tuning fully supports CoT demonstrations and is not limited to explicit $(x,y)$ pairs.** To evaluate Context Tuning, we follow the standard few-shot $(x,y)$ setup from Prefix Tuning, TTT, and MetaICL. Tasks where $y$ labels contain CoT reasoning tokens fit directly into this format, so Context Tuning naturally supports it. In addition, Context Tuning is not fundamentally limited to explicit input-output pairs: Context Tuning takes in tokens that represent the demonstration context and optimizes it, with the leave-one-out objective updating this representation by masking parts of the context in a similar spirit to masked language modeling. Therefore, Context Tuning can be applied to more free-form contexts even though our evaluation uses the conventional $(x,y)$ setup for compatibility with prior works.
> >
> > **Context Tuning addresses the limitations of zero-shot and instruction-driven prompting.** Zero-shot prompting is inherently constrained because LLM systems routinely face out-of-distribution tasks and the need for personalization, where pretrained behavior alone is insufficient. Instruction-driven prompting may achieve comparable performance, but it requires significant manual effort in designing task-specific prompts. In contrast to both, Context Tuning adapts to test-time tasks from demonstrations without manual prompt engineering.
> >
> > **Few-shot adaptation is natural in real-world interactive systems and open-ended scenarios**, where organizations often possess small amounts of labeled examples for new tasks/domains/clients, and specifying desired input-to-output behavior through examples is often more reliable than relying on a single instruction. Context-Tuning is designed for this setting by achieving robust few-shot adaptation performance across demonstration counts (Figure 4a), label noise (Figure 4b), and model sizes (Table 2).
> >
> > ### **Concluding Note**
> >
> > Given the acknowledged clarity of the presentation and the consistent gains of Context Tuning over baselines, along with the explanations on novelty, model-task pairing, and generality provided here, we kindly ask you to consider raising the score, as we believe the method offers practical and broadly useful contributions to the community.

---

### Official Review · Reviewer_Tm2d · 2025-11-04

**Soundness:** 2
**Presentation:** 2
**Contribution:** 1
**Rating:** 0
**Confidence:** 4

**Summary:**

This paper proposes Context Tuning, a method that improves few shot adaptation (with additional parameters) of LLMs without updating model weights. The core idea is to initializing existing prompt tuning and prefix tuning method with embeddings obtained directly from task specific demonstration examples. As a result, the paper introduces two variants: CT Prompt tunes a soft prompt initialized from the demonstrations. CT KV initializes and tunes layerwise key value prefixes extracted from the model activations on the same demonstrations. The authors further introduce leave one out masking to prevent target leakage during training, and token dropout to reduce overfitting. Across multiple benchmarks, Context Tuning outperforms standard prompt and prefix tuning and is competitive with test time training while being substantially more efficient, especially for CT KV.

**Strengths:**

- Parameter efficiency and stability. The method keeps model weights frozen and only learns a small set of prompt or prefix parameters, which preserves the base model while enabling task adaptation.

- Consistent gains in few shot settings. Experiments show improvements over vanilla in context learning and over standard prompt or prefix tuning on several benchmarks. The gains are robust across a range of demonstration counts and across different base model sizes. But, much of the empirical benefit appears to come from a stronger initialization of existing prompt or prefix tuning, rather than a fundamentally new optimization mechanism. I list this more fully under weaknesses.

**Weaknesses:**

- Limited novelty in mechanism. The main algorithmic move is to initialize the trainable prompt or prefix from demonstration embeddings, then optimize as in standard prompt or prefix tuning. This is a strong and practical idea, but conceptually close to prior methods and may be seen as an improved initialization strategy rather than a new optimization principle.

- Positioning relative to test time training. The paper reports results for TTT plus CT KV and presents this combined setting among the proposed methods. Since TTT is an existing and separate approach that updates model weights, it is not clear that TTT plus CT KV should be considered an integral contribution of this paper. A clearer separation of baselines, proposed methods, and combinations would help, along with ablations that isolate where the gains come from.

- Interpretation of robustness to label noise. The paper highlights that CT KV remains strong even when many demonstration labels are corrupted. However, in an in context optimization view, a method should ideally adjust its behavior when labels change. If performance is largely unchanged under heavy label corruption, this can suggest that the method relies more on input side regularities or priors than on learning from the provided labels. This reduces the strength of the robustness claim as evidence of learning from context.

- Missing specification details. Some core definitions and operations are underspecified. In leave one out masking, the construction of P^{\text{minus i}}_{\text{CT}} for prompts and of the corresponding context representation for prefixes is not formally detailed. The paper says the relevant tokens are masked out from the attention view, but it does not specify whether this is implemented by zeroing keys and values, by blocking attention through an attention mask, or by removing those prefix positions from the cache. A precise definition would improve clarity and reproducibility.

**Questions:**

- Variable length to fixed size initialization. Demonstration contexts can vary in length. How do you map variable length demonstrations to a fixed number of trainable prompt tokens or to a fixed per layer prefix budget. For CT Prompt, do you truncate or pool embeddings when the demonstration context exceeds the prompt length. For CT KV, how are per layer prefix lengths chosen and how are multiple demonstrations combined or allocated across that budget.

---

> ### Author Response · Authors · 2025-11-21
>
> We thank the reviewer for the thoughtful and detailed feedback. We are encouraged that the reviewer recognizes our method’s consistent gains over strong baselines while keeping model weights frozen. We thoroughly address the concerns below and have updated the PDF accordingly.
>
> > Limited novelty in mechanism.
>
> The reviewer correctly pointed out that Context Tuning involves an improved initialization strategy. However, we motivate and design two changes, Leave-One-Out Masking and Token Dropout, to the optimization loop in Section 4.3. In Table 3, we present an ablation study over all datasets that demonstrates the importance of both design decisions to Context Tuning’s performance. Without them, a naive use of the new initialization strategy (the “Neither” row in Table 3) performs even worse than Prefix Tuning on NLP-LR and BBH (Table 1). Therefore, Leave-One-Out Masking and Token Dropout are not auxiliary tricks added to squeeze out extra gains; they are core components that make Context Tuning effective.
>
> > Positioning relative to test time training… A clearer separation of baselines, proposed methods, and combinations would help…
>
> We agree that TTT is a separate existing method and point this out in Section 4.2, where we also formalize TTT as an instance of our In-Context Optimization framework. In Table 1, we ablate the performance of TTT (under “Baselines”), CT-KV, and TTT+CT-KV separately over all datasets, matching the reviewer’s request. In the revised PDF, we made a minor change by putting TTT+CT-KV under “Combined Methods” to avoid confusion.
>
> This ablation reveals two insights:
> 1. **CT-KV matches TTT’s accuracy at a fraction of optimization time.** We additionally derive the time complexity for both methods and show that CT-KV is more efficient because it does not recompute the KV-cache corresponding to the demonstration examples, while TTT does so for each training iteration.
> 2. **TTT and CT-KV are complementary.** TTT+CT-KV achieves higher performance than when TTT or CT-KV is used separately, indicating that optimizing weights (TTT) and context (CT-KV) can be effectively combined to achieve the best performance.
>
> > Interpretation of robustness to label noise.
>
> We agree that a method should adjust its behavior when labels change, and observe it in various experiments: in Figure 4b, CT-KV’s accuracy indeed decreases at higher corruption rates, showing that it uses label semantics rather than ignoring them. Moreover, Figure 4a shows that CT-KV benefits strongly from far more demonstration pairs than the number of unique multiple-choice labels. If CT-KV relied primarily on input-side regularities or priors, these effects would not appear consistently across datasets.
>
> Our robustness experiment targets a **complementary question**: does the method catastrophically fail once some labels are corrupted? Figure 4b shows that up to 75% corruption, CT-KV still outperforms Prefix Tuning and ICL, indicating that it learns a robust task representation under imperfect demonstration quality, an important property when real-world demonstration examples may include inconsistencies or user errors.
>
> > Missing specification details.
>
> We appreciate the reviewer pointing this out. In Section 4.3, $P_{\mathrm{CT}}^{-i}$ ​is constructed by removing the key and value vectors in $P_{\mathrm{CT}}$ that correspond to the in-context demonstration pair $(x_i, y_i)$. At the implementation level, we achieve this by blocking attention through an attention mask (setting the corresponding positions of the mask to 0). We have revised Section 4.3 in the updated PDF to clarify this.
>
> To ensure reproducibility, we additionally provide experiment setup and hyperparameters in Section 5.3 and Appendix D, submitted our code in the Supplementary Materials, and will open-source a public implementation upon acceptance.
>
> > Variable length to fixed size initialization.
>
> We clarify that both CT-Prompt and CT-KV use the full, variable-length contexts directly, without truncation or mapping. Therefore, the number of trainable prompt tokens in CT-Prompt and the number of trainable key or value vectors per layer in CT-KV equals the number of tokens in the demonstrations. Since we designed and evaluated Context Tuning for few-shot adaptation, CT-KV significantly outperforms Prefix Tuning with only small extra runtimes (Table 1).
>
> ### **Concluding Note**
>
> We appreciate your engagement and hope you will consider updating the score to reflect the substantial amount of work we put into designing, analyzing, and validating Context Tuning, as well as the meaningful empirical and methodological contributions it offers. Given your acknowledgement of Context Tuning’s consistent gains over baselines and our strengthened explanations here, we believe the paper provides value to the community much beyond what the current score would suggest.

---

### Author Response · Authors · 2025-12-04

We thank all reviewers for their time, thoughtful feedback, and constructive suggestions. We are encouraged that reviewers highlighted our paper’s clear presentation (HFNj, Qgpc), consistent improvements over baselines (HFNj, Tm2d), and comprehensive experiments (Qgpc).

We provided detailed responses to each reviewer and revised the manuscript as follows.
- We updated Table 1 to clearly separate TTT, CT-KV, and TTT+CT-KV under **Baselines**, **Our Methods**, and **Combined Methods** headers respectively. This clear separation allows a clear understanding of Context Tuning’s effectiveness, either used alone or with TTT.
- To improve reproducibility, we revised Section 4.3 to state explicitly that Leave-One-Out Masking is implemented by setting the attention mask to 0 at the positions corresponding to the masked demonstration pair.

To address questions regarding novelty, we clarified that Context Tuning introduces algorithmic improvements beyond the new initialization strategy (Tm2d), and we emphasized its conceptual and implementation-level distinctions from existing baselines (HFNj, Qgpc).

Additionally, we responded to reviewer Qgpc’s thoughtful question on the positioning of test-time optimization relative to In-Context Learning. Specifically, we clarify that we view In-Context Optimization (ICO) as a bridge between ICL and finetuning that can outperform either traditional method alone, and that while task-specific prompt design requires substantial human effort, Context Tuning offers automated, task-agnostic improvements at test time.

We thank the reviewers again for their valuable feedback and insightful questions.

---

### Meta-Review · Area_Chair_PGjy · 2025-12-15

**Summary:**

The paper aims at improving the performance of few-shot adaptation of LLM without fine-tuning. Their methods is built upon LLLM's in-context learning ability and propose a method that initializes trainable vectors from the few-shot examples of a novel task, then optimizes them to
solve each example.

Strengths identified by reviewers include the paper's presentation and its efficiency in performance and computational costs.

However, the paper also faces several critical weaknesses that have not been addressed in the rebuttal. All reviewers share the same concern that the proposed method that initializes the trainable prompt or prefix from demonstration embeddings, then optimizes as in standard prompt or prefix tuning, which is very close to prior methods and is incremental. The proposed method has limited generality and applicability across different tasks and applications scenarios such as math, coding, COT etc while general capability to handle multiple tasks simultaneously within a single model is the true merit of LLMs.

Given these considerations, this paper clearly fails to meet the standards of ICLR. The authors are encouraged to address the highlighted issues to strengthen their contribution to the field of few shot adaptation LLM.

**Reviewer Concerns:**

The authors provided detailed feedback and most of concerns have been addressed, including a) clearer clarification and comparisons to TTT; b) Interpretation of robustness to label noise; c) Missing specification details; d) Variable length or fixed size initialization; e) the need of clearer explanations of each designs' contributions.

However, two main concerns are still not fully addressed: 1) the proposed method that initializes the trainable prompt or prefix from demonstration embeddings, then optimizes as in standard prompt or prefix tuning, is very close to prior methods and is incremental; 2) the proposed method has limited generality and applicability across different tasks and applications scenarios such as math, coding, COT etc.

**Reviewer Scores:**

The initial reviewer scores are negative (one strong reject, one reject and one borderline reject). Reviewer Tm2d would have considered increasing the score but all reviewers would have maintained negative scores as two main concerns are not fully addressed.

---

### Decision · Program_Chairs · 2026-01-26

Reject